# Missing rating imputation based on product reviews via deep latent variable models

Dingge Liang [* 1]   Marco Corneli [* 2 1]   Pierre Latouche [* 3]   Charles Bouveyron [* 1]

## Abstract

We introduce a deep latent recommender system (deepLTRS) for imputing missing ratings based on the observed ratings *and* product reviews. Our approach extends a standard variational autoencoder architecture associated with deep latent variable models in order to account for both the ordinal entries and the text entered by users to score and review products. DeepLTRS assumes a latent representation of both users and products, allowing a natural visualisation of the positioning of users in relation to products. Numerical experiments on simulated and real-world data sets demonstrate that DeepLTRS outperforms the state-of-the-art, in particular in contexts of extreme data sparsity.

## 1. Introduction and related works

Matrix completion is a central machine learning problem, which consists in predicting the non observed entries of a matrix on the basis of the observed entries. We focus here on the collaborative filtering problem which aims at completing a matrix of user ratings about products. These matrices are extremely sparse in practice which makes the inference of the non-observed entries challenging. A long series of approaches have been proposed to tackle this issue.

The works of Gopalan et al. (2015); Basbug & Engelhardt (2016) rely on the assumption that ratings are sampled from Poisson distributions, simple or compound. Recently, coupled compound Poisson factorization (CCPF, Basbug & Engelhardt, 2017) was introduced as a more general framework based on different generative approaches (e.g. mixture

models or matrix factorization models). Another class of collaborative filtering models rely on user-based (respectively item-based) autoencoders to produce user (item) embeddings in lower dimension, both based on recurrent (Monti et al., 2017) or convolutional (Sedhain et al., 2015; Zheng et al., 2016) architectures. The last two approaches are special cases of a more general architecture (graph autoencoder models, Kipf & Welling, 2016), recently employed for matrix completion (Berg et al., 2017).

Unfortunately, although the aforementioned models can account for side information additionally to the user ratings, they do not introduce a modeling framework specific to the product reviews. However in practice, the product ratings are often paired with reviews that might contain crucial information about the user preferences. Thus, in McAuley & Leskovec (2013), the hidden factors and hidden topics (HFT) model combines latent rating factors with latent review topics. Still, when a large amount of user ratings is missing, the performance of the predictions turns out to be limited.

We introduce here the deep latent recommender system (deepLTRS, Sections 2-3) for the completion of rating matrices, accounting for the textual information collected in the product reviews. DeepLTRS extends probabilistic matrix factorization (Mnih & Salakhutdinov, 2008) by relying partially on latent Dirichlet allocation (LDA, Blei et al. (2003)) and its recent autoencoding extensions (Srivastava & Sutton, 2017; Dieng et al., 2019). Thus, our approach adopts a variational autoencoder architecture as a generative deep latent variable model for both the ordinal matrix encoding the user/product scores, and the document-term matrix encoding the reviews. Our approach is tested on simulated and real datasets (Section 4) and compared with other state-of-the-art approaches in contexts of extreme data sparsity.

## 2. A text based recommender system

We consider data sets involving $M$ users scoring/reviewing $P$ products. Such data sets can be encoded by two matrices: an ordinal data matrix $Y$ accounting for the *scores* that users assign to products and a document-term matrix (DTM) $W$ accounting for the *reviews* that users make about products.

---

*Equal contribution  [1]Université Côte d'Azur, Inria, CNRS, Laboratoire J.A.Dieudonneé, Maasai team, Nice, France  [2]Université Côte d'Azur, Center of Modelling, Simulation and Interactions (MSI), Nice, France  [3]Université de Paris, Laboratoire MAP5, Paris, France. Correspondence to: Marco Corneli <marco.corneli@univ-cotedazur.fr>.

*Presented at the first Workshop on the Art of Learning with Missing Values (Artemiss) hosted by the $37^{th}$ International Conference on Machine Learning (ICML).* Copyright 2020 by the author(s).

**Ordinal data.** The ordinal data matrix $Y$ in $\mathbb{N}^{M \times P}$ is such that $Y_{ij}$ is the score that the $i$-th user assigns to the $j$-th product. This matrix can be very sparse (most of its entries are missing) corresponding to users *not* scoring/reviewing some products. Conversely, when a score is assigned it takes values in $\{1, \ldots, H\}$ with $H > 1$.

**Assumption 1.** *Henceforth, we assume that an ordinal scale is consistently defined. For instance, when customers evaluate products,* 1 *always means "very poor" and* $H$ *is always associated with "excellent" reviews.*

**Assumption 2.** *The number of ordered levels* $H$ *is assumed to be the same for all (not missing)* $Y_{ij}$. *If it is not the case, a scale conversion pre-processing algorithm (see for instance Gilula et al., 2018) can be employed to normalize the number of levels.*

**Text data.** By considering all the available reviews, it is possible to store all the different vocables employed by the users into a *dictionary* of size $V$. Thenceforth, we denote by $W^{(i,j)}$ a row vector of size $V$ encoding the review by the $i$-th user to the $j$-th product. The $v$-th entry of $W^{(i,j)}$, denoted by $W_v^{(i,j)}$, is the number of times (possibly zero) that the word $v$ of the dictionary appears into the corresponding review. The **document-term matrix** $W$ is obtained by row concatenation of all the row vectors $W^{(i,j)}$.

**Assumption 3.** *For the sake of clarity, we assume that the review* $W^{(i,j)}$ *exists if and only if* $Y_{ij}$ *is observed.*

Note that, since each row in $W$ corresponds to one (and only one) not missing entry in $Y$, the number of rows in the DTM is the same as the number of observed values in $Y$.

It is now assumed that both users and products have latent representations in a low-dimensional space $\mathbb{R}^D$, with $D \ll \min\{M, P\}$. In the following, $R_i$ denotes the latent representation of the $i$-th user. Similarly $C_j$ is the latent representation of the $j$-th product.

The following generative model is now considered:

$$Y_{ij} = \langle R_i, C_j \rangle + \epsilon_{ij}, \forall i = 1, ..., M, \forall j = 1, ..., P, \quad (1)$$

where $\langle \cdot, \cdot \rangle$ is the standard scalar product and the residuals $\epsilon_{ij}$ are assumed to be i.i.d. and normally distributed random variables, with zero mean and unknown variance $\eta^2$:

$$\epsilon_{ij} \sim \mathcal{N}(0, \eta^2).$$

In the following, $R_i$ and $C_j$ are seen as random vectors, such that

$$\begin{aligned} R_i &\overset{\text{i.i.d}}{\sim} \mathcal{N}(0, I_D), \forall i \\ C_j &\overset{\text{i.i.d}}{\sim} \mathcal{N}(0, I_D), \forall j \end{aligned} \quad (2)$$

with $R_i \perp\!\!\!\perp C_j$. This model is knows as probabilistic matrix factorization (PMF, Mnih & Salakhutdinov, 2008). Note

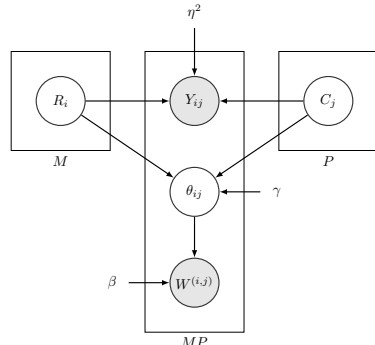

*Figure 1.* Graphical representation of the generative model (variational parameters are not included).

that, due to rotational invariance of PMF, the choice of isotropic prior distributions for $R_i$ and $C_j$ is in no way restrictive

We now extend PMF by also relying on $R_i$ and $C_j$ to characterize the document-term matrix $W$. Following the generative model of LDA (Blei et al., 2003), each document $W^{(i,j)}$ is drawn from a mixture distribution over a set of $K$ latent topics. The topic proportions in the document $W^{(i,j)}$ are denoted by $\theta_{ij}$, a vector lying in the $K - 1$ simplex. LDA, in its multinomial PCA formulation assumes that

$$p(W^{(i,j)}|\theta_{ij}) \sim \text{Multinomial}(L_{ij}, \beta\theta_{ij}), \quad (3)$$

where $L_{ij}$ is the number of words in the review $W^{(i,j)}$ and $\beta \in \mathbb{R}^{V \times K}$ is the matrix whose entry $\beta_{vk}$ is the probability that vocable $v$ occurs in topic $k$. By construction, $\sum_{v=1}^V \beta_{vk} = 1, \forall k$. Moreover, conditionally to all vectors $\theta_{ij}$, all the reviews $\{W^{(i,j)}\}$ are independent random vectors. The following assumption calls $R_i$ and $C_j$ into play.

**Assumption 4.** *The topic proportions are now assumed to be sampled as follows*

$$\theta_{ij} = \sigma(f_\gamma(R_i, C_j)), \quad (4)$$

*where* $f_\gamma : \mathbb{R}^{2D} \to \mathbb{R}^K$ *is a continuous function approximated by a neural network parametrized by* $\gamma$ *and where* $\sigma(\cdot)$ *denotes the softmax function.*

We emphasize that the $\theta_{ij}$ are no longer independent contrary to LDA. We finally state that:

**Assumption 5.** *Given the pair* $(R_i, C_j)$, *it is assumed that* $Y_{ij}$ *and* $W^{(i,j)}$ *are independent.*

We stress that we are *not* assuming that $Y_{ij}$ and $W^{(i,j)}$ are independent. Instead, we describe a framework in which the dependence between them is completely captured by the latent embedding vectors $R_i$ and $C_j$. A graphical representation of the generative model described so far can be seen in Figure 1.

## 3. Variational auto-encoding inference

A natural inference procedure associated with the generative model proposed would consist in looking for estimates $(\eta^2, \gamma, \beta)$ maximizing the (integrated) log-likelihood of the observed data $(Y, W)$. Unfortunately, this quantity is not directly tractable and we rely on a variational lower bound (a.k.a. ELBO) to approximate it. A tractable family of joint distributions $q(\cdot)$ over the pair $(R, C)$ of all $(R_i)_i$ and $(C_j)_j$ is considered via the following *mean-field* assumption

$$q(R, C) = q(R)q(C) = \prod_{i=1}^{M} \prod_{j=1}^{P} q(R_i)q(C_j). \quad (5)$$

Moreover, since $R_i$ and $C_j$ follow Gaussian prior distributions (Eq. (2)), $q(\cdot)$ is assumed to be as follows:

$$q(R_i) = g(R_i; \mu_i^R, S_i^R), \quad (6)$$
$$q(C_j) = g(C_j; \mu_j^C, S_j^C), \quad (7)$$

where $g(\cdot; \mu, S)$ is the pdf of a Gaussian multivariate distribution with mean $\mu$ and variance $S$ and

$$\mu_i^R := h_{1,\phi}(Y_i, W^{(i,\cdot)}) \qquad S_i^R := h_{2,\phi}(Y_i, W^{(i,\cdot)})$$
$$\mu_j^C := l_{1,\iota}(Y^j, W^{(\cdot,j)}), \qquad S_j^C := l_{2,\iota}(Y^j, W^{(\cdot,j)}).$$

Here, $Y_i$ (respectively $Y^j$) denotes the $i$-th row (column) of $Y$ and $W^{(i,\cdot)} := \sum_j W^{(i,j)}$ corresponds to a document concatenating all the reviews written by user $i$ and, similarly $W^{(\cdot,j)} := \sum_i W^{(i,j)}$ corresponds to all the reviews about the $j$-th product. The two functions $h_\phi : \mathbb{R}^{P+V} \to \mathbb{R}^{2 \times D}$ and $l_\iota : \mathbb{R}^{M+V} \to \mathbb{R}^{2 \times D}$ are the network *encoders* and they are parametrized by $\phi$ and $\iota$, respectively.

Thanks to Eqs. (1)-(3)-(5)-(6)-(7) and by computing the KL divergences between $q(\cdot)$ and the prior distributions of $R$ and $C$, the *evidence lower bound* (ELBO) can be explicitly computed (*cf.* Section A, supplementary material).

## 4. Numerical experiments

### 4.1. Simulated data and effect of the data sparsity

**Simulation setup.** An ordinal data matrix $Y$ with $M = 750$ rows and $P = 600$ columns is simulated according to a latent continuous cluster model. The rows and columns of $Y$ are randomly assigned to two latent groups, in equal proportions. Then, for each pair $(i, j)$ corresponding to an entry of $Y$, a Gaussian random variable $Z_{ij}$ is sampled in such a way that $Z_{ij} \sim \mathcal{N}(2, 1)$ if $X_i^{(R)} = X_j^{(C)}$, and $Z_{ij} \sim \mathcal{N}(3, 1)$ otherwise, where $X_i^{(R)}$ and $X_j^{(C)}$ label the clusters of the $i$-th row and the $j$-th column, respectively. Then the following thresholds $t_0 = -\infty, \quad t_1 = 1.5, \quad t_2 = 2.5, \quad t_3 = 3.5, \quad t_4 = +\infty$ are used to sample the note

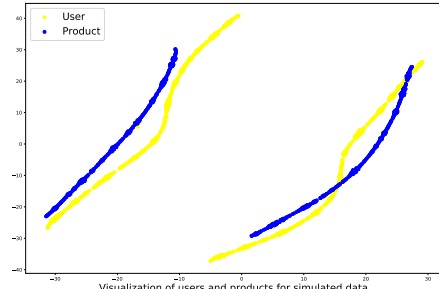

*Figure 2.* Visualization of the user and product embeddings (sparsity of 0.99).

$Y_{ij} \in \{1, ..., 4\}$ as

$$Y_{ij} = \sum_{k=1}^{4} k \mathbf{1}(Z_{ij})_{]t_{k-1}, t_k[} \quad (8)$$

Then, four different texts from the BBC news (denoted by A, B, C, D) are used to build a message associated to the note $Y_{ij}$ according to the scheme summarized in Table 1.

|  | cluster 1 | cluster 2 |
|---|---|---|
| cluster 1 | A | B |
| cluster 2 | C | D |

*Table 1.* Topic assignments for simulated data.

Thus, when the user $i$ in cluster $X_i^{(R)} = 2$ rates the product $j$ in cluster $X_j^{(C)} = 1$, a random variable $Z_{ij} \sim \mathcal{N}(3, 1)$ is sampled, $Y_{ij}$ is obtained via Eq. (8) and the review $W^{(i,j)}$ is built by random extraction of words from message C. All the sampled messages have an average length of 100 words. Finally and in order to introduce some noise, only 80% of words are extracted from the main topics, while the remaining 20% is extracted from the other topics at random.

First, Figure 2 shows a t-SNE representation of $R_i$ and $C_j$, with data sparsity of 0.99 (i.e. 99% of the observations in $Y$ and $W$ are replaced by NA at random). We first note that the two (row ans column) clusters are well separated despite the large degree of sparsity. Since deepLTRS assumes that the closer the distance, the greater the probability that the product is reviewed by the user, this latent representation is well representative of the simulated setup . A total of 10 data sets was simulated according to the above setup, with sparsity rates varying in the interval $[0.5, 0.99]$. The whole observed data is used as *training* data set, the remaining missing data was split into 50% for *validation* and 50% for *test*. This experimental setup was used to benchmark deepLTRS by comparing it to some state-of-the-art methods as HFT (McAuley & Leskovec, 2013), HPF (Gopalan et al., 2015) and CCPF (Basbug & Engelhardt, 2017). Since

CCPF has many choices of combination between sparsity and response models, we chose one example with better performance as described in (Basbug & Engelhardt, 2017). Figure 3 shows the evolution of the test RMSE of deepLTRS (with $D = 50$ and text) and its competitors. Additional results are reported in the supplementary material, Section C. Let us recall that the simulation setup does not follow the deepLTRS generative model and therefore does not favour any method here.

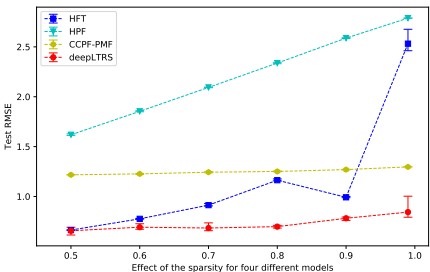

*Figure 3.* Test RMSE of models: HFT, HPF, CCPF and deepLTRS with different sparsity level on the simulated data.

## 4.2. Amazon Fine Food data

**Data and pre-processing.** This data set[1] spans over a period of more than 10 years, including all $568,464$ reviews up to October 2012. All records include product and user information, ratings, time of the review and a plain-text review. In the data pre-processing step, we only considered users with more than 20 reviews and products reviewed by more than 50 users to obtain more meaningful information. The retained data was processed by removing all punctuations and numbers. Since around half of the negative reviews have more positive than negative words in the data set, we kept the stop words (such as "not","very") to make our bag-of-words structure more clearly retain the original semantics. The final data set has $M = 1,643$ users, $P = 1,733$ products, a vocabulary with $V = 5,733$ unique words and $32,811$ text reviews in total. The data sparsity is here of 0.989%.

**Rating prediction.** Five independent runs of the algorithm were performed. For each run, we randomly selected 80% of the non-missing data as the training set, 10% for validation and the remaining 10% for testing. Table 2 reports the test RMSE for deepLTRS and its competitors (HFT, HPF and CCPF-PMF) on the predicted ratings for the Amazon Fine Food data. Once again, deepLTRS has better performance than other models, with an average test RMSE equal to 1.2518. In order to deeper understand the latent representation meaning, we provide in Figure 4 a t-SNE

---

[1]The data set can be downloaded freely at `https://snap.stanford.edu/data/web-FineFoods.html`

*Table 2.* Test RMSE on Amazon Fine Food data.

| Model | Run 1 | Run 2 | Run 3 | Runt 4 | Run 5 | Average |
|---|---|---|---|---|---|---|
| HFT | 1.424 | 1.533 | 1.474 | 1.423 | 1.385 | 1.448 ($\pm$0.051) |
| HPF | 2.949 | 2.968 | 2.931 | 2.943 | 2.973 | 2.953 ($\pm$0.016) |
| CCPF-PMF | 1.269 | 1.296 | 1.303 | 1.292 | 1.295 | 1.291 ($\pm$0.011) |
| deepLTRS | **1.136** | **1.259** | **1.244** | **1.171** | **1.247** | **1.252** ($\pm$0.049) |

visualisation ($D = 50$) of the user latent positions on two specific latent variables (var. 3 and 11) that can be easily interpreted according to average ratings (top) and number of reviews (bottom) the users give to the products. Indeed, it clearly appears that var. 11 captures the rating scale of Amazon users whereas var. 3 seems to encode the user activity (number of reviews). Additional analyses are reported in the supplementary material, Section D.

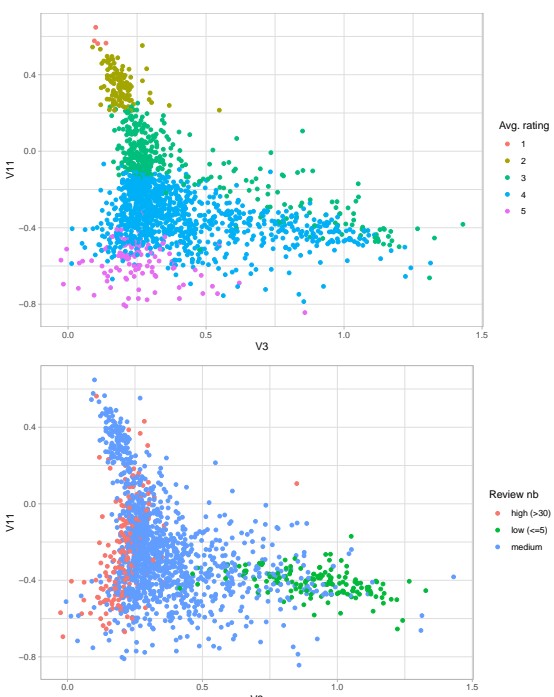

*Figure 4.* Latent representation of users on var. 3 and 11, according to average ratings (left) and numbers of reviews (right) they give to products.

## 5. Conclusion

We introduced here the deepLTRS model for ratings imputation using both the ordinal and the text data available. Our approach adopts a variational autoencoder architecture as a generative deep latent variable model for both the ordinal matrix encoding the user/product scores, and the document-term matrix encoding the reviews. The further ability of deepLTRS to predict the most likely words used by a reviewer to review a product will be inspected in future works.

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

# Supplementary material

## A. Evidence lower bound

$$
\begin{aligned}
\text{ELBO}(\Theta) = &\sum_{i,j} \left( \mathbb{E}_{q(R_i,C_j)} \left[ -\frac{1}{2} \left( \frac{(Y_{ij} - (R_i^T C_j))^2}{\eta^2} + \log \eta^2 \right) \right] \right) \\
&+ \sum_{i,j} \left( \mathbb{E}_{q(R_i,C_j)} \left[ \left( W^{(i,j)} \right)^T \log \left( \beta \sigma(f_\gamma(R_i, C_j)) \right) \right] \right) \\
&- \sum_i \left[ -\frac{1}{2} \left( tr(S_i^R) + (\mu_i^R)^T \mu_i^R - D - \log |S_i^R| \right) \right] \\
&- \sum_j \left[ -\frac{1}{2} \left( tr(S_j^C) + (\mu_j^C)^T \mu_j^C - D - \log |S_j^C| \right) \right] + \xi
\end{aligned}
\tag{9}
$$

where $\Theta := \{\eta^2, \gamma, \beta, \phi, \iota\}$ denotes the set of the model and variational parameters and $\xi$ is a constant term that includes all the elements not depending on $\Theta$.

## B. Architecture of deepLTRS

In the architecture of deepLTRS, we have two encoders for users and products separately. As a method for stochastic optimization, we adopt an Adam optimizer, with learning rate $lr = 2e^{-3}$.

In the user encoder, the first hidden layer has $init\_dim\_R = (P + V)$ neurons, where $P$ is equal to the number of products, and $V$ is the number of words in the text vocabulary; the second hidden layer has $mid\_dim = 50$ neurons. In the product encoder, the first hidden layer has $init\_dim\_C = (M + V)$ neurons, where $M$ is equal to the number of users; the second hidden layer has the same neurons as in the user encoder. $Softplus$ activation function and batch normalization are applied in each layer.

In the decoder, the first two layers have $2 \times int\_dim$ and 80 neurons separately, where $int\_dim = init\_dim\_R$ when decoding for users and $int\_dim = init\_dim\_C$ for products. The number of neurons in the third layer depends on the number of topics, here we used $nb\_of\_topics = 50$. In addition, $Relu$ activation function and batch normalization are applied. In order to obtain the probability of each word, the $Softmax$ function is used in the end.

## C. More on simulated data experiments

**DeepLTRS with and without text data.** We first run a simulation to highlight the interest in using the reviews to make more accurate predictions of the ratings. To do so, 10 data sets are simulated according to the above simulation setup, with sparsity rates varying in the interval $[0.5, 0.99]$. Figure 5 shows the evolution of the test RMSE of deepLTRS (with $D = 50$), with and without using text data, versus the data sparsity level. One can observe that, even though both models suffer the high data sparsity (increasing RMSE), the use of the text greatly help deepLTRS to maintain a high prediction accuracy for data sets with many missing values. Furthermore, the use of text reviews greatly reduce the variance of the deepLTRS predictions.

## D. More on Amazon Fine Food data experiments

Figure 6 shows a visualization with t-SNE of the high-dimensional latent representations (here $D = 50$) of the users and products for the Amazon data. The density of the overlapping regions and the distance between user and product embeddings reflect the probability of users reviewing the corresponding products.

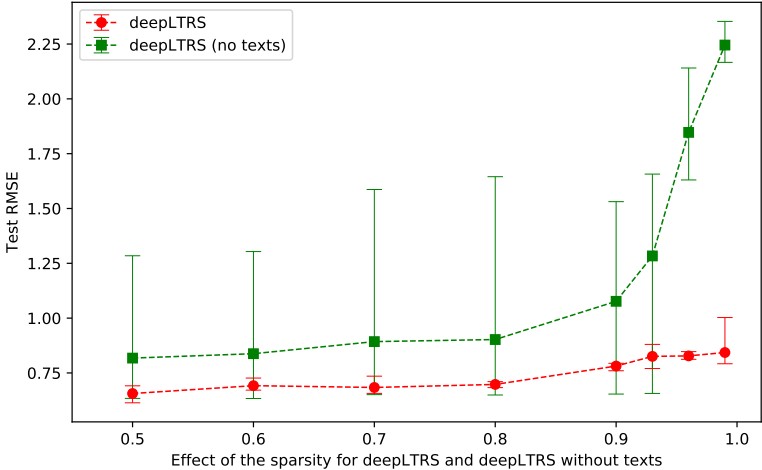

*Figure 5.* RMSE by deepLTRS with and without text information on simulated data.

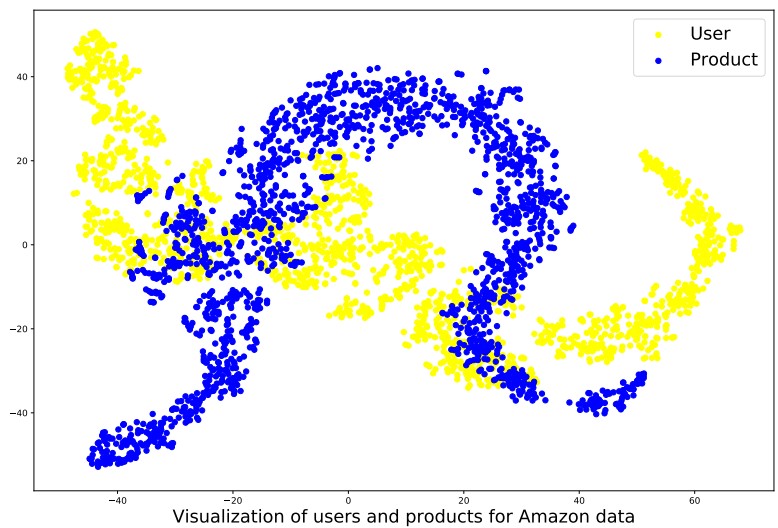

*Figure 6.* Projection with t-SNE of user and product latent representations for the Amazon Fine Food data set.

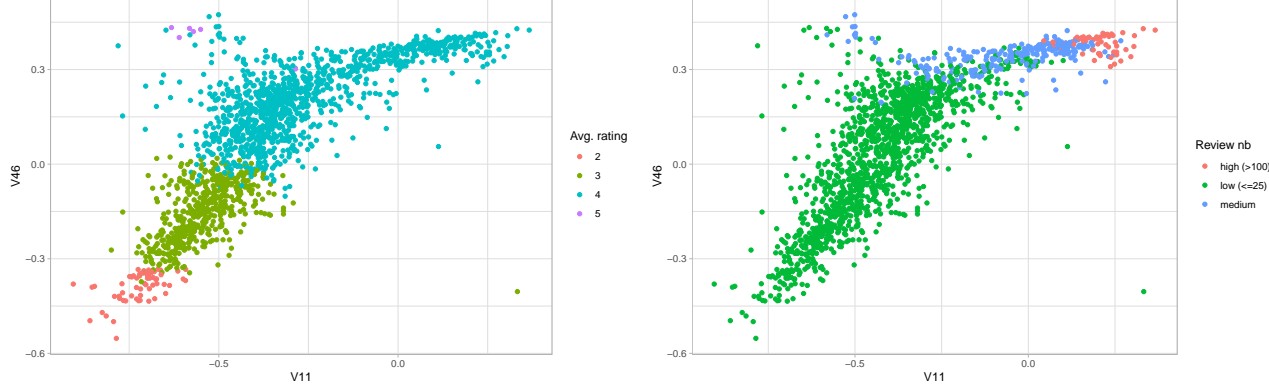

*Figure 7.* Latent representation of products on var. 11 and 46, according to average rating (left) and number of reviews (right) they receive from users.