# OpenReview forum: "Missing rating imputation based on product reviews via deep latent variable models"
_ICML.cc/2020/Workshop/Artemiss — ICML Artemiss 2020_

### Official Review · AnonReviewer1 · 2020-06-23
**Handling sparsity in product review data with VAE**

**Confidence:** 3
**Rating:** 7

**Review:**

The description of adjusted VAE is unclear to me. Do you estimate ($\eta$, $\lambda$, $\beta$) as well as R and C distributions' parameters in the inference network?  What is the conditional distribution over the observed rating given latent variables?
In equation 6 and 7, I think $S$ is $\Sigma$.

In the first experiment, you used the whole observed data for training and the remaining missing data for test and validation. In the second experiment, you randomly picked data for training without mentioning the process of handling sparsity during training the VAE.

The missing error in the CCPF method doesn't increase with the growing missing-rate, which is puzzling.

---

### Decision · Program_Chairs · 2020-07-02

**Decision:**

Accept

**Comment:**

We are very happy to inform you that your paper has been accepted for the Artemiss workshop. We will contact you soon to inform you about the details concerning the format of your presentation at the workshop, and the camera-ready version deadline. Please take into account the referee's comments to write the camera-ready version.